# Gelatin Methacryloyl (GelMA) Hydrogel Scaffolds: Predicting Physical Properties Using an Experimental Design Approach

**DOI:** 10.3390/ijms241713359

**Published:** 2023-08-29

**Authors:** Corentin Peyret, Kamil Elkhoury, Sabine Bouguet-Bonnet, Sophie Poinsignon, Corentin Boulogne, Tristan Giraud, Loïc Stefan, Yasmina Tahri, Laura Sanchez-Gonzalez, Michel Linder, Ali Tamayol, Cyril J.F. Kahn, Elmira Arab-Tehrany

**Affiliations:** 1Université de Lorraine, LIBio, F-54000 Nancy, France; 2Université de Lorraine, CNRS, CRM2, F-54000 Nancy, France; 3Université de Lorraine, CNRS, LCPM, F-54000 Nancy, France; 4Department of Biomedical Engineering, University of Connecticut Health Center, Farmington, CT 06030, USA

**Keywords:** gelatin methacryloyl, experimental design, multiscale characterisation, NMR, rheology, compression

## Abstract

There is a growing interest for complex in vitro environments that closely mimic the extracellular matrix and allow cells to grow in microenvironments that are closer to the one in vivo. Protein-based matrices and especially hydrogels can answer this need, thanks to their similarity with the cell microenvironment and their ease of customization. In this study, an experimental design was conducted to study the influence of synthesis parameters on the physical properties of gelatin methacryloyl (GelMA). Temperature, ratio of methacrylic anhydride over gelatin, rate of addition, and stirring speed of the reaction were studied using a Doehlert matrix. Their impact on the following parameters was analyzed: degree of substitution, mass swelling ratio, storage modulus (log(G’)), and compression modulus. This study highlights that the most impactful parameter was the ratio of methacrylic anhydride over gelatin. Although, temperature affected the degree of substitution, and methacrylic anhydride addition flow rate impacted the gel’s physical properties, namely, its storage modulus and compression modulus. Moreover, this experimental design proposed a theoretical model that described the variation of GelMA’s physical characteristics as a function of synthesis conditions.

## 1. Introduction

Inside human tissues, cells interact within a complex three-dimensional (3D) microenvironment while spreading, migrating, proliferating, and differentiating in the extracellular matrix (ECM) [1]. Nowadays, most of the commonly used models are two-dimensional (2D) cell cultures [2,3], which misrepresents the structural and functional characteristics of native tissues. More realistic 3D models that can actually mimic these characteristics need to be developed because they will allow cells to interact with ECM components and adjacent cells in all directions, and will provide more accurate cellular behavior and functions [4].

Due to their high-water retention ability and resemblance to the ECMs, hydrogels have been extensively used to create such realistic 3D in vitro models [5,6,7,8]. Hydrogels can be formed from many polymers, such as collagen, the most abundant protein in the human body and the principal constituent of skin, bone, and connective tissues [9]. The partial hydrolysis of native collagen produces gelatin, that in turn can be crosslinked to form biodegradable and biocompatible hydrogels [10]. Moreover, gelatin contains promotors of cell remodeling such as target sequences of matrix metalloproteinase (MMP) and cell attachment enhancers such as arginine-glycine-aspartic acid (RGD) sequences [11].

Chemical modification of gelatin prior to crosslinking allows the bypass of some of its limitations, such as its low mechanical modulus and its rapid degradation [12]. One example that has gained immense popularity in tissue engineering, regenerative medicine, and drug delivery [13,14,15,16,17] is gelatin methacryloyl (GelMA) because it is stable at 37 °C and mechanically stronger than unmodified gelatin hydrogels [13,18]. GelMA requires the substitution of methacrylate groups on the amine-containing side groups of gelatin [19,20,21]. This methacrylation reaction allows, in the presence of a photoinitiator, the UV-light induced polymerization of gelatin.

Controlling GelMA’s physical properties is important in tissue engineering applications, since cells can feel and respond to the stiffness of their surroundings [22]. For example, the elasticity of soft matrices has been shown to direct stem cell lineage specification [23]. Although GelMA has been widely used as a biomaterial, the influence of the methacrylation reaction parameters on the physical properties of GelMA has mostly been studied by varying only one factor at a time, which does not take into consideration the interaction between set parameters and their effect on GelMA properties. In this paper, a design of experiment methodology was chosen to study the influence of synthesis parameters and their interactions on the physical properties of GelMA. The experimental design approach made it possible to vary multiple parameters at the same time, study their impact on the different responses, establish an empirical model to predict the behavior of GelMA, and finally, to perform an ANOVA assessing the statistical relevance of the data.

The aim of the present work was to fully understand the effect and interactions of the most common parameters used to control the methacrylation reaction of GelMA, and consequently, to control and study the physical properties of the resulting crosslinked hydrogels. Several key parameters for GelMA synthesis were identified and used for optimization, namely, temperature, ratio of methacrylic anhydride over gelatin, methacrylic anhydride addition flow rate, and stirring speeds. The experimental design led to a quadratic empirical model that was used to control the degree of substitution and physical properties of GelMA.

## 2. Results and Discussion

### 2.1. Hydrogel Crosslinking Conditions

To modulate the physical responses of GelMA, it is possible to change the dilution factor of GelMA. It has been shown that more concentrated GelMA will be stiffer and more resistant to enzyme degradation [24]. It is also worth noting that others have shown the importance of UV irradiation on the different physical parameters of the crosslinked hydrogel [25]. Specifically, GelMA that has been cured for longer periods of time will be stiffer, swell less, and degrade at a slower rate [26,27,28,29]. Also, there is evidence that gelation of GelMA at 4 °C before curing has an impact on the molecular weight between two bonds, especially with low DoS gels, and therefore the rheological properties [30]. Hence, curing conditions in this study are a factor to consider when comparing these results to the rest of the literature.

### 2.2. NMR Analysis for GelMA Degree of Substitution and Synthesis Conditions 

Gelatin analyzed with ^1^H NMR displays a complex spectrum due to the presence of many amino acid proton peaks. Therefore, in order to quantify the lysine substitution rate, the peak area of the aromatic amino-acid region (δ = 7.23–7.5 ppm) of gelatin was used for normalization between the different samples. The peak area of lysine methylene protons, located between δ = 3–3.2 ppm, between unmodified gelatin and GelMA, was compared to calculate the DoS. Spectra are displayed in Figure 1A.

In the literature, the GelMA synthesis temperature is mostly between 40 and 60 °C. For example, Van den Bulcke et al. used a reaction temperature of 50 °C [31]; therefore, a range of temperature between 40 and 60 °C was explored in this study. However, MA was allowed to react only 2 h instead of the more commonly found 3 h reaction time. This could explain the difference observed in the lysine substitution rate in these results when compared to similar reaction conditions, but with different reaction durations [32]. 

The quantity of methacrylic anhydride was the main focus when varying the synthesis parameters in other studies, because it is understood to be the main factor for the DoS [12,33]. Although this study leans in that direction, it was demonstrated that other synthesis parameters played a role in fine tuning the physical properties of the final gel. Finally, stirring speed is a parameter that has rarely been explored. And consequently, there was rarely any precision on the actual stirring speed in other studies. The reaction solution is described as being “gently stirred”, “vigorously stirred”, or “continuously stirred”, or otherwise unspecified [26,34,35]. This work brought an original approach on this matter by integrating stirring speed as one of the independent variables.

### 2.3. Surface Response Study

The experimental design results are summarized in Table 1. For each experiment, the following hydrogel characteristics were analyzed: lysine substitution rate (Y1), swelling (Y2), storage modulus (Y3), and compression modulus (Y4). Table 2 gives the coefficients of linear, quadratic, and interaction parts of the equation. The numbers in bold specify the most significant regression index for each response; coefficients of determination (R²) are also highlighted.

For the regression indices describing linear terms of the equation, the temperature applied during synthesis had a negative impact on swelling and a positive impact on compression at 15%, considering that regression indices have a value of −0.712 and 57.852, respectively. However, the [MA]/[Gelatin] ratio had a positive impact on both the lysine substitution rate and the log(G’) with regression indices equal to 26.622 and 0.376, respectively. The MA flow rate and stirring did not appear to have a significant impact on any of the responses.

Regarding the coefficients describing the quadratic terms, the regression index for the [MA]/[Gelatin] ratio was the most significant regardless of the response. Regression indices displayed a negative impact of the [MA]/[Gelatin] ratio on itself for the lysine substitution rate, as well as for the log(G’) and compression at 15% with a value of −31.995, −1.497, and −97.782, respectively. Nevertheless, the coefficient was positive for swelling, exposing that the [MA]/[Gelatin] ratio has an agonistic effect on itself with a value of 4.235.

Finally, the interaction between temperature and the [MA]/[Gelatin] ratio had a positive impact on swelling and a negative impact on the log(G’) since regression indices had a value of 10.88 and −2.124, respectively. Conversely, the interaction between the [MA]/[Gelatin] ratio and the MA flow rate had a negative impact on the lysine substitution rate and compression at 15% since regression indices were −10.861 and −100.062, respectively.

### 2.4. Effect of Experimental Parameters on Lysine Substitution Rate

The first response to be studied was the lysine substitution rate; 24 experiments were conducted and the results are shown in Table 1.

The model consistency deduced from the Doehlert experimental design and its fitness values were expressed by the coefficient of determination (R² = 0.896 for Y1), which was greater than 0.800 for this response and indicated that the polynomial model fitted well with the experimental data. The ANOVA showed the great level of significance both for the coefficient of determination and repeatability; they are situated over a 99.99% confidence interval.

As mentioned above, the coefficients of each term of the quadratic model describing the lysine substitution rate response can be found in the second column of Table 2. The most influential parameter was the [MA]/[Gelatin] ratio, which had an agonistic effect on the response, i.e., this parameter had a positive effect on the response. Whereas for the quadratic terms, it was a strong antagonistic effect for high values; that is, the [MA]/[Gelatin] ratio had an inhibition effect on itself.

Finally, the interaction terms of the model revealed a strong antagonistic effect between the [MA]/[Gelatin] ratio and the addition rate of MA, suggesting that when both parameters were at their maximum value, the response diminished. As shown in Figure 2A, the MA flow rate needed to be brought down in order to maximize the lysine substitution rate. The interaction between the [MA]/[Gelatin] ratio and the stirring speed presented an antagonistic effect, implying that the overall molecular agitation ought to be minimized in order to augment the substitution rate of lysine; this could be due to the lack of accessibility of the lysine sites by the methacrylic acid as well as the diminishing pH as the concentration of MA augmented in the reaction medium. In order to reach a DoS up to 80%, the PBS buffer without pH adjustment was sufficient [12,36,37,38], but for higher substitution rate, other studies have found that adjusting the pH during or before the reaction allowed higher substitution rates, up to 40% more when comparing the same quantity of MA [39,40]. Furthermore, pH-adjusted carbonated buffer permitted to reach a 100% DoS, even having methacryloyl groups on carboxylate amino-acids [40,41,42]. 

The fact that the medium molecular agitation should be reduced is supported by the optimization path shown in Figure 2B,C. In Figure 2B, the substitution rate of gelatin has an exponential progression and it appeared to slow down when the [MA]/[Gelatin] ratio became higher than 77.5%. However, this implied that the extremely high substitution of gelatin can be attained with a very high [MA]/[Gelatin] ratio. Moreover, in Figure 2C, the lysine substitution rate can be improved by decreasing the stirring speed to 450 rpm. On the other hand, to minimize the response, the only parameter that was required to be lowered was the [MA]/[Gelatin] ratio.

We showed that the DoS for gelatin was mainly dependent on the quantity of methacrylic anhydride added to the solution, and, to a lesser extent, on the reaction temperature and stirring speed. Lowering these two parameters, in addition to maximizing the [MA]/[Gelatin] ratio, yielded highly substituted hydrogels. 

### 2.5. Effect of Experimental Parameters on Swelling of the Gel

The swelling of the gel was measured to assess its capacity to absorb water; results are shown in Table 1.

The correlation of experimental data with the empirical model was given by the coefficient of determination, with a value R² = 0.949, meaning the empirical model fitted very well with the experimental data. The ANOVA analysis also showed that the model accurately describes the effects of the experimental parameters on the gel’s swelling, with the coefficient being in the >99.99% confidence interval. The equation coefficients taken from Table 2 show that the linear terms of the equation did not play a significant part in the model. For the quadratic terms, the coefficient b_2-2_ was the highest, with a strong agonistic effect, underlying the importance of the [MA]/[Gelatin] ratio to the response.

The swelling parameter was often associated with the substitution parameter; Nichol et al. exposed an inverted correlation between the substitution rate and the swelling ratio, i.e., the higher the substitution rate, the lower the swelling ratio [12]; other studies have confirmed this [10,40]. This work correlated this observation, showing that the DoS around 12% had a swelling ratio of 19, while the DoS around 71% had a swelling ratio of 6.8. Finally, for the interaction terms of the equation, the strongest interaction was between the temperature and the [MA]/[Gelatin] ratio. Figure 3A exhibits that this surface response had a saddle-like shape, signifying that when both variables were either high or low, the response was maximized; whereas, when one of the variables was minimized, the response was minimal as well. The optimization path in Figure 3B,C reveals that in order to minimize the response, the only parameter that needed to be lowered was the [MA]/[Gelatin] ratio, whereas all the others were higher than their central value. On the other hand, swelling can be maximized by reducing the [MA]/[Gelatin] ratio as well as the temperature.

This looked to be in line with the previous response; since swelling reflected the capacity of the polymer to retain water, it was strongly dependent on the gelatin substitution rate. However, the empirical model indicated that for a temperature of 55 °C and a [MA]/[Gelatin] ratio of 120%, the swelling ratio will be high, around 13. This effect could be due to temperature having an inhibitory effect on the condensation reaction between methacrylic acid and lysine, resulting in a looser gelatin network once polymerized.

### 2.6. Effect of Experimental Parameters on the Storage Modulus (log(G’))

Rheological properties of hydrogels are important to assess their capability to withstand shear stress. Hence, the storage modulus was measured, and the results are shown in Table 1. The storage modulus was taken as the decimal logarithm of the average of the viscoelastic domain linear part, as illustrated in Figure 1B. This set of data had a correlation coefficient of R² = 0.878, which is greater than 0.800, indicating that an empirical polynomial model is relevant to describe the dataset. The ANOVA analysis showed a good level of confidence, with the correlation coefficient being in the >99.99% confidence interval. The regression indices describing the model can be found in Table 2.

These coefficients showed that the linear terms of the equation did not impact the model. Regarding the quadratic terms, the most important parameter was once again the [MA]/[Gelatin] ratio, with an antagonistic effect. When both independent variables were high, the storage modulus decreased, and when one of the variables was at a minimum value, the storage modulus increased (Figure 4A). Similarly, when the [MA]/[Gelatin] ratio was just below the central value and the temperature was high, the gel was more rigid. Interestingly, the gels were most rigid when the [MA]/[Gelatin] ratio was high and when the temperature was around 55 °C and the [MA]/[Gelatin] ratio around 60%. 

Figure 4B,C showed the optimization path for the response. The rigidity augmented very rapidly until the central value and then rose again (Figure 4B). For the optimization path of each parameter separately, Figure 4C indicates that in order to minimize the response, the [MA]/[Gelatin] ratio needs to be minimized as well as the temperature, and the flow rate of the MA needs to be augmented slightly. On the other hand, to maximize the storage modulus, both the temperature and the stirring speed need to be lowered, and the [MA]/[Gelatin] ratio must be augmented. As expected, the rigidity of the hydrogel was linked to the DoS; this was coherent with previous reports [10,31]. But the fact that the average DoS GelMA, around 45% DoS according to the model, would be so rigid was new.

### 2.7. Effect of Experimental Parameters on the Compression Modulus at 15%

The compression modulus was measured to test the capacity of the gel to deform under constraint. It was measured at 15% compression, corresponding to the slope of the linear region of the stress-strain curve, as is displayed in Figure 1C; the results are shown in Table 1. 

The correlation coefficient had a value of R² = 0.72 with a relevance >99.99%, which was not greater than 0.8, suggesting the quadratic empirical model may not be the most relevant model to describe the variation of the compression modulus. However, what the terms of the equation indicated, was that for the linear terms of the equation, the temperature was the most influential factor, with a strong agonistic effect and also the most relevant one. For the quadratic terms of the equation, the strongest factor was b_2-2_, followed shortly by b_1-1_. The b_2-2_ factor being negative meant that the [MA]/[Gelatin] ratio had an antagonistic effect on itself. The b_1-1_ factor being positive meant that the temperature had an agonistic effect on itself. Finally, for the interaction terms of the equation, the most significant interaction was an antagonistic effect between the [MA]/[Gelatin] ratio and the addition rate of MA, denoting that the two parameters had an inhibitory effect on each other.

The surface response for this interaction is shown in Figure 5A. Because the interaction was negative when both parameters were high, the elastic modulus was low, and when one parameter was minimized, the response was augmented. To maximize the compression modulus, a high [MA]/[Gelatin] ratio and a low MA flow rate were needed, confirming the fact that the compression modulus was dependent on the DoS of GelMA and therefore on the [MA]/[Gelatin] ratio. This trend has already been reported previously [43,44]. Figure 5A also illustrates that the maximum value for this interaction is when the [MA]/[Gelatin] ratio is high and the MA flow rate is low. 

Figure 5B,C shows that in order to minimize the response, the addition rate of MA needs to be minimized, and the stirring speed needs to be brought down to half of its value. However, the only parameter that was required to be augmented to maximize the elastic modulus was the temperature, while all the other parameters remained at their central value. Remarkably, gels show a very low value for the compression modulus, of around 100 Pa, but it has been reported that the compressive modulus can greatly vary from one GelMA to the next, from 700 Pa [39] for low DoS gels, up to 55 kPa [34,45] for high DoS gels.

## 3. Materials and Methods

### 3.1. Reagents

Gelatin from porcine skin (Type A, 300 bloom), methacrylic anhydride (MA), Irgacure 2959 (PI) (2-hydroxy-4′-(2-hydroxyethoxy)-2-methylpropiophenone), collagenase from Clostridium histolyticum, and Dulbecco’s phosphate-buffered saline (DPBS) were purchased from Sigma-Aldrich (Saint-Quentin-Fallavier, France). Deuterium oxide (99.9 atom % D) was purchased from Eurisotop (Saint-Aubin, France).

### 3.2. GelMA 

#### 3.2.1. GelMA Synthesis

GelMA with different substitution degrees was synthesized according to the method adopted by Van den Bulcke et al. [31]. Briefly, gelatin was dissolved at 10% in DPBS at 50 °C under stirring. Methacrylic anhydride was added in the adequate proportion with a syringe pump (KD scientific, Holliston, MA, USA), and the mixture was allowed to react for 2 h at controlled temperatures and stirring velocity, following the experimental design. After 2 h, the reaction was stopped by the addition of 3 times the volume of warm (50 °C) DPBS. To remove the excess of unreacted methacrylic acid, the mixture was dialyzed for 5 days at 50 °C against distilled water using 12–14 kDa cut-off dialysis tubing. The solution was then freeze dried for at least 48 h, and the recovered white porous foam was stored at −20 °C until further use. 

#### 3.2.2. Hydrogel Preparation

Hydrogels were prepared by dissolving 10% (*w*/*v*) of freeze-dried GelMA in DPBS at 50 °C. Photoinitiator was added at 1% (*w*/*v*) and dissolved under stirring at 50 °C. The GelMA solution was then crosslinked using a CL-1000L ultraviolet crosslinker (UVP, Cambridge, UK) for 4 min in a circular silicone mold of 20 mm diameter and 2 mm height.

### 3.3. Determination of GelMA Degree of Substitution

The GelMA degree of substitution (DoS) was determined by proton nuclear magnetic resonance. Spectra were collected at 40 °C and 400 MHz (^1^H resonance frequency) using a Bruker Avance III 400 MHz instrument (Bruker, Karlsruhe, Germany). Then, 50 mg of the GelMA sample was dissolved in 1 mL of deuterium oxide at 50 °C. Spectra were acquired (64 accumulations) with a relaxation delay of 7 s in order to ensure that the measurement was quantitative; the residual water signal was eliminated by using presaturation. NMR spectra were phased and the baseline corrected before measuring the peak area of the region of interest. DoS was calculated with the following equation:(1)$$DS=\left(1-\frac{Area\;lysine\left[GelMA\right]}{Area\;lysine\left[Gelatin\right]}\right)*100$$

### 3.4. Rheological Testing

Dynamic rheological measurements were performed using an AR2000 Rheometer (TA Instruments, New Castle, DE, USA) operating in oscillatory mode and equipped with a Peltier plate temperature control and a 20.0 mm diameter parallel plate. Stress sweep experiments were carried out in the range of 0.46–2000 Pa applied stress (10 points per decade) at a frequency of 1 Hz with a constant normal force applied (0.4 N), and at a temperature of 37 °C. All the measurements were repeated three times. 

### 3.5. Mechanical Testing

The compression tests were performed using a universal mechanical testing machine (Lloyd-LRX, Lloyd Instruments, Fareham, UK). Polymerized hydrogels disks were compressed at a crosshead speed of 10 mm/s until fracture occurred. Prior to all measurements, the zero gap was determined, and samples were compressed three times at 5% deformation. Each condition was tested four times. 

### 3.6. Mass Swelling Ratio

The mass swelling ratio was measured using 3 samples. The samples were immersed in DPBS at 37 °C for 24 h. Then the samples were removed from the DPBS, and excess water was gently removed using a paper tissue. The swollen weight was measured using a precision scale. Afterward, the samples were freeze dried for at least 48 h, and the dry weight of each sample was measured. The mass swelling ratio was then calculated using the following formula:(2)$$Mass\;swelling\;ratio=\frac{Swollen\;weight\;of\;the\;sample}{Dry\;weight\;of\;the\;sample}$$

### 3.7. Experimental Design

An experimental design was based on a preliminary screening design (not disclosed here) in order to evaluate the effect of the four main parameters on the structural and mechanical properties of GelMA. The reaction conditions are detailed in Table 3. A Doehlert experimental design was used to study the influence of processing variables on the structural and physical properties of GelMA.

Temperature (X1), methacrylic anhydride over gelatin ratio (X2), methacrylic anhydride flow rate (X3), and stirring speed (X4) were selected as independent variables. The methacrylic anhydride over gelatin ratio was defined as the following:(3)$$\frac{\left[MA\right]}{\left[Gelatin\right]}=\frac{volume\;of\;MA\;(mL)}{mass\;of\;gelatin\;(g)}$$

The experimental matrix (Table 1) was constructed using NEMRODW software (NEMRODW, version 2017, Marseille, France). The Doehlert matrix offers a uniform distribution of experimental points within the experimental domain and a good flexibility. In a Doehlert design, the number of experiments is equal to K² + K + cp, where K is the number of independent variables, and cp is the number of repetitions of the central point. A total of 24 experiments were performed, including the triplicate of the center point.

The lysine substitution rate (Y1) was used to measure the DoS of gelatin. The swelling ratio (Y2) was tested in order to have information on the physical properties of the gel. Finally, the storage modulus (Y3), expressed as log(G’), and compression modulus at 15% (Y4) were measured to provide information about the mechanical behavior of the gel. Using experimental data, a model was created; it uses a quadratic equation to approximate each response:(4)$$Y=b_{0}+\sum_{i=1}^{k}b_{i}x_{i}+\sum_{i=1}^{k}b_{ii}x_{i}^{2}+\sum_{i=1}^{k-1}\sum_{j=i+1}^{k}b_{ij}x_{i}x_{j}$$
where Y is the studied dependent response, where the i and j vary from 1 to 4 (number of independent variables studied here). b_0_ and b_i_ are, respectively, the constant term and the regression coefficient for linear effects; b_ii_ are regression indices for squared effects, and b_ij_ are regression indices for interaction effects. x_j_ and x_j_ are coded experimental levels of the independent variables.

### 3.8. Statistics

Analysis of variance (ANOVA), regression, and graphical study were examined using NEMRODW software (NEMRODW, version 2017, Marseille, France).

## 4. Conclusions

In this study, the experimental design method was used to investigate the influence of four parameters of GelMA synthesis and how they affected the physical and mechanical properties of the resulting hydrogels. 

This work corroborates the fact that the DoS is mainly dependent on the quantity of methacrylic anhydride added to the solution, and highlights that a low reaction temperature and a low stirring speed favor a high DoS.

The swelling capability of the gel depended on both the reaction temperature and the [MA]/[Gelatin] ratio with an agonistic interaction, i.e., both parameters needed to be either maximized or minimized in order to have a high swelling ratio. Similarly, the storage modulus also depended on the reaction temperature and the [MA]/[Gelatin] ratio, but with an antagonistic interaction, so that when both parameters were either maximized or minimized, the storage modulus was low. Finally, for the compression modulus, there was a strong antagonistic interaction between the [MA]/[Gelatin] ratio and the flow rate of MA. This implied that, in order to either maximize or minimize the response, both parameters needed to be at their minimum value. Interestingly, the maximum value was reached when all the parameters were at their central value, with the notable exception of the temperature, which needed to be maximized.

Overall, this study confirmed the importance of the [MA]/[Gelatin] ratio as a parameter for the physical properties, but it highlighted the importance of all the factors related to the stirring of the solution, i.e., the addition rate of MA, the stirring speed, and the temperature. Particularly, the empirical model determined by the experimental design permitted a hydrogel with tunable physical properties within the limits of the experimental domain.

## Figures and Tables

**Figure 1 ijms-24-13359-f001:**
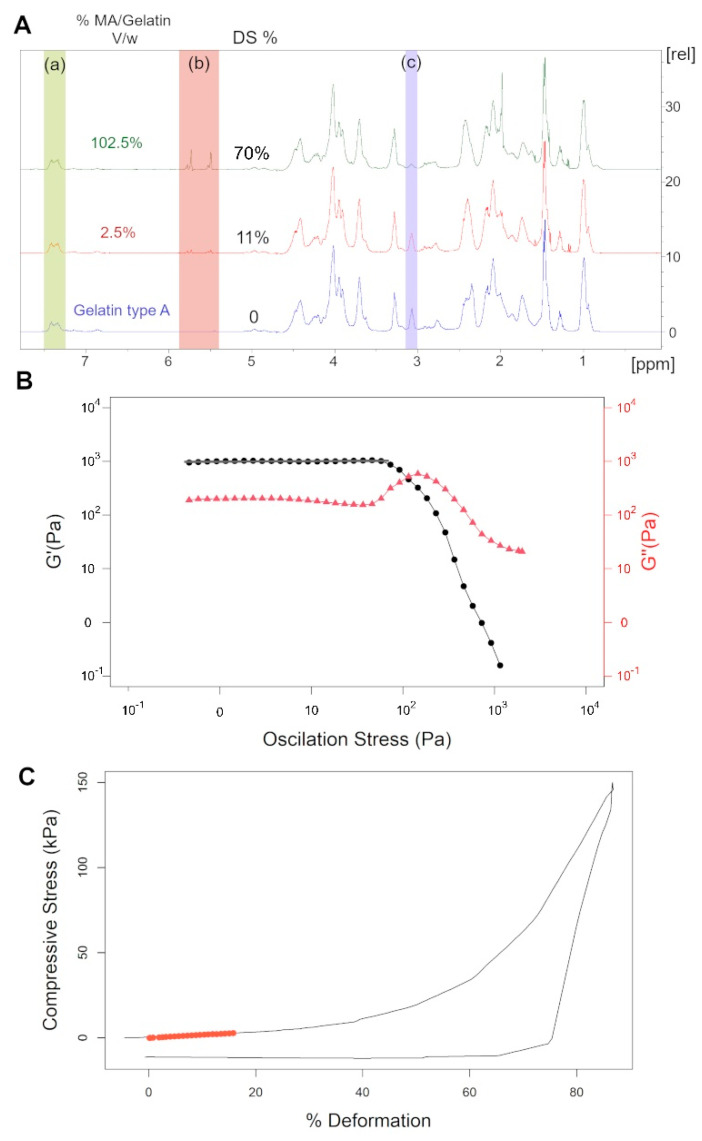
Spectra and diagrams used for the data analysis. (**A**) ^1^H NMR spectra of samples 13, 4, and unmodified gelatin, respectively. (a) 7.23–7.5 ppm: green peak area corresponding to aromatic amino-acid region and were used as reference. (b) 5.39–5.85 ppm: red peak area of methylene protons; these peaks indicate the presence of methacryloyl groups (c) 3–3.2 ppm: blue peak area of lysine methylene protons; these peaks were used to calculate the degree of substitution. (**B**) Representative diagram of sample 12 rheological profile; the storage modulus was taken as the average of linear viscoelastic domain, indicated with the grey line. (**C**) Diagram of the compression test of hydrogels; the red circles indicate the slope taken to determine the compression modulus.

**Figure 2 ijms-24-13359-f002:**
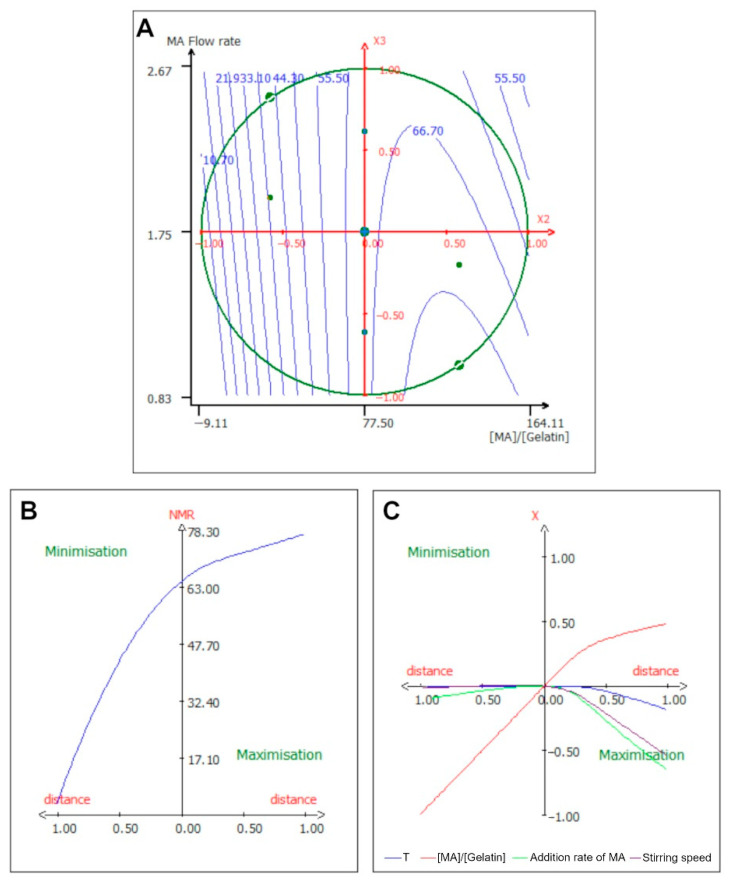
Isocurve of interaction parameters and optimization pathway for the lysine substitution rate. (**A**) Surface plot showing the effects of the mutual interaction between the [MA]/[Gelatin] ratio (X2) and MA flow rate (X3) on the lysine substitution rate, while the other variables are kept at their center point. (**B**) Optimization path of the response in order to maximize or minimize the lysine substitution rate; each independent variable has the same variation in centered reduced value. (**C**) Optimization path of the response (*X*-axis) for each separate parameter (*Y*-axis) expressed in reduced value and centered reduced value, respectively.

**Figure 3 ijms-24-13359-f003:**
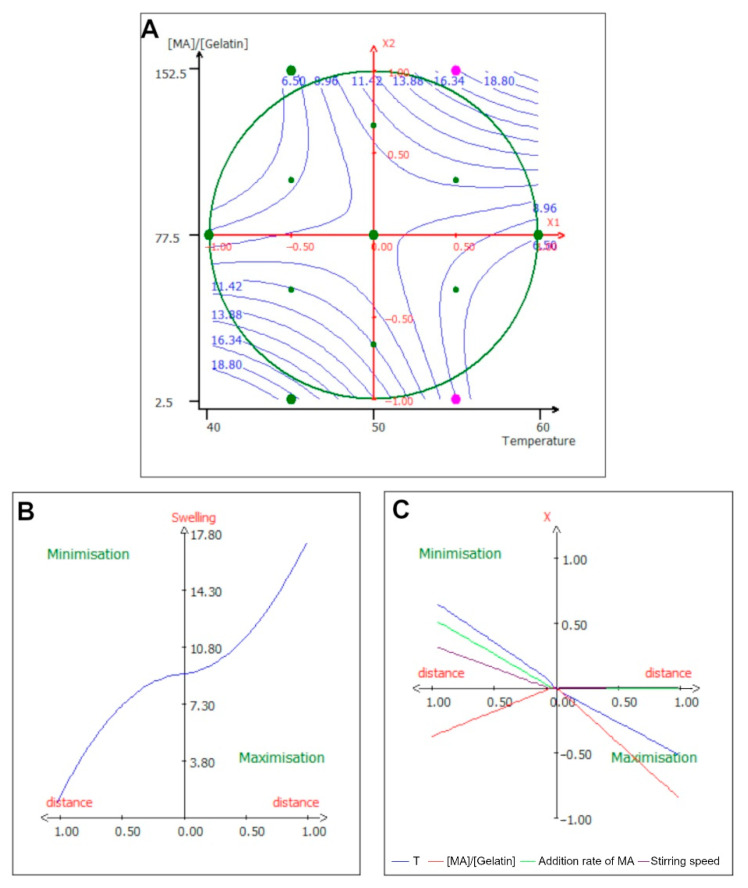
Isocurve of interaction parameters and optimization pathway for the swelling ratio. (**A**) Surface plot showing the effects of the mutual interaction between the temperature and the [MA]/[Gelatin] ratio on the swelling ratio, while the other variables are kept at their center point. (**B**) Optimization path of the response in order to maximize or minimize the swelling ratio; each independent variable has the same variation in centered reduced value. (**C**) Optimization path of the response (*X*-axis) for each separate parameter (*Y*-axis) expressed in centered reduced value.

**Figure 4 ijms-24-13359-f004:**
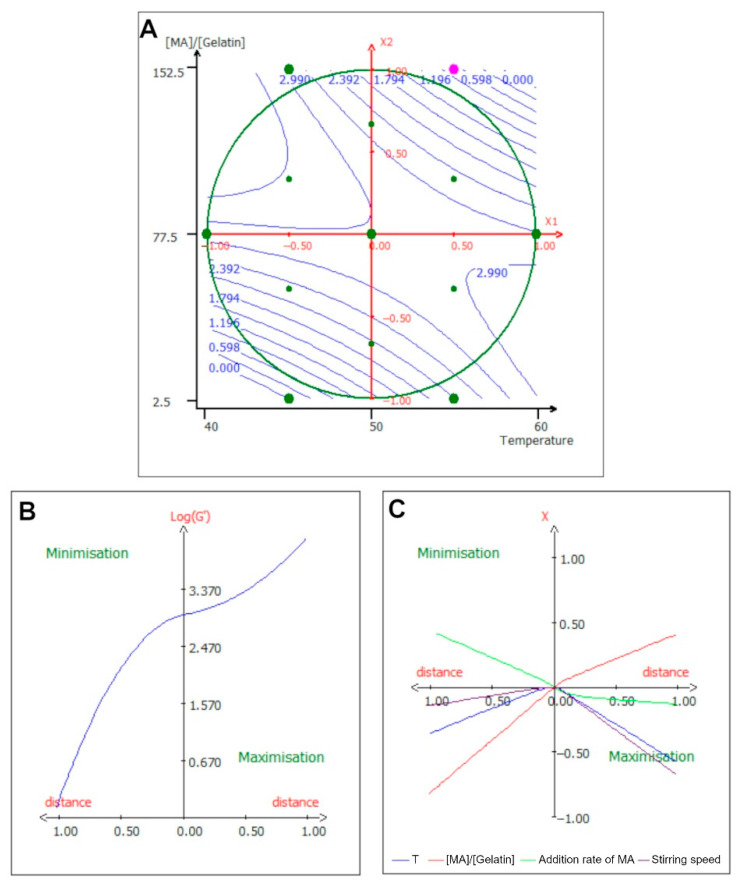
Isocurve of interaction parameters and optimization pathway for log(G’). (**A**) Surface plot showing the effects of the mutual interaction between the temperature and [MA]/[Gelatin] ratio on log(G’), while the other variables are kept at their center point. (**B**) Optimization path of the response in order to maximize or minimize log(G’); each independent variable has the same variation in centered reduced value. (**C**) Optimization path of the response (*X*-axis) for each separate parameter (*Y*-axis) expressed in centered reduced value.

**Figure 5 ijms-24-13359-f005:**
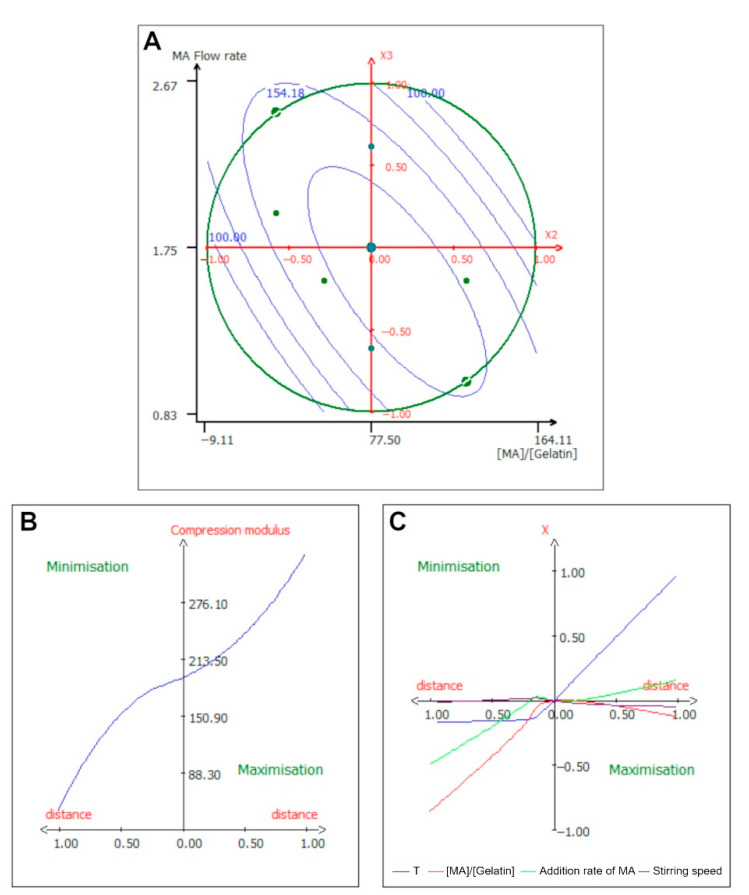
Isocurve of interaction parameters and optimization pathway for the compression modulus at 15%. (**A**) Surface plot showing the effects of the mutual interaction between the temperature and the [MA]/[Gelatin] ratio on the compression modulus at 15%, while the other variables are kept at their center point. (**B**) Optimization path of the response in order to maximize or minimize the compression modulus at 15%; each independent variable has the same variation in centered reduced value. (**C**) Optimization path of the response (*X*-axis) for each separate parameter (*Y*-axis) expressed in centered reduced value.

**Table 1 ijms-24-13359-t001:** Doehlert experimental matrix and experimental data obtained by applying the experimental design.

Exp n°	Independent Variables				Responses			
	X1 T	X2 [MA]/[Gelatin]	X3 Addition Rate of MA	X4 Stirring Speed	Y1 % Lysine Substitution	Y2 Swelling	Y3 Log(G’)	Y4 Compressive Modulus 15%
	°C	mL/g	mL/min	rpm			Pa	Pa
1	60	77.5%	1.75	550	54.18	7.14	3.05	313
2	40	77.5%	1.75	550	71.76	7.55	2.41	204
3	55	152.5%	1.75	550	67.92	6.98	3.12	155
4	45	2.5%	1.75	550	12.33	19.28	-	-
5	55	2.5%	1.75	550	12.11	-	1.75	-
6	45	152.5%	1.75	550	68.65	7.29	3.21	174
7	55	102.5%	2.5	550	67.47	7.62	2.27	156
8	45	52.5%	1.0	550	50.75	8.05	3.19	147
9	55	52.5%	1.0	550	58.06	7.65	3.21	139
10	50	127.5%	1.0	550	70.79	6.78	1.81	188
11	45	102.5%	2.5	550	58.22	6.90	2.81	110
12	50	27.5%	2.5	550	46.68	8.19	2.67	196
13	55	102.5%	2.0	650	70.35	7.59	2.84	199
14	45	52.5%	1.5	450	52.86	9.87	2.91	130
15	55	52.5%	1.5	450	60.72	6.58	2.71	272
16	50	127.5%	1.5	450	68.80	8.00	2.97	183
17	50	77.5%	2.25	450	64.98	8.39	2.93	153
18	45	102.5%	2.0	650	56.84	9.49	3.00	126
19	50	27.5%	2.0	650	45.77	8.52	2.93	130
20	50	77.5%	1.25	650	61.11	8.67	2.36	155
21	50	77.5%	1.75	550	63.75	9.09	2.96	174
22	50	77.5%	1.75	550	64.99	8.73	2.96	184
23	50	77.5%	1.75	550	63.71	9.05	2.98	181
24	50	77.5%	1.75	550	66.26	9.72	2.97	251

**Table 2 ijms-24-13359-t002:** Coefficients of quadratic models for each parameter, taken from NEMRODW software (NEMRODW, version 2017, Marseille, France).

Name	% Lysine Substitution	Swelling	Log (G’)	Compressive Modulus 15%
b_0_	64.674 ***	9.148 ***	2.965 ***	193.636 ***
b_1_	0.182	−0.712 ***	−0.043	57.852 ***
b_2_	26.622 ***	−0.552 ***	0.376 ***	7.098
b_3_	−1.089 ***	0.050	−0.301 ***	−7.586
b_4_	−2.102 ***	0.227 *	−0.064 **	−9.142
b_1-1_	−1.704 ***	−1.799 ***	−0.206 ***	73.530 ***
b_2-2_	−31.995 ***	4.235 ***	−1.497 ***	−97.782 ***
b_3-3_	−0.594	−3.393 ***	−0.495 ***	−59.137 ***
b_4-4_	−0.33	−1.36 ***	0.32 ***	−36.369 *
b_1-2_	−0.293	10.88 ***	−2.124 ***	−44.094
b_1-3_	1.296 *	−3.76 ***	0.521 ***	50.292 *
b_1-4_	3.348 ***	−2.789 ***	0.786 ***	−11.735
b_2-3_	−10.861 ***	2.37 ***	1.656 ***	−110.062 ***
b_2-4_	−4.012 ***	1.931 ***	−1.094 ***	−26.538
b_3-4_	2.472 ***	−3.087 ***	0.718 ***	25.251
R²	0.896 ***	0.947 ***	0.878 ***	0.720 ***

*** = relevance > 99.99% (for 0.0001 < *p* value < 0.001) ** = relevance > 99% (for 0.001 < *p* value < 0.01) * = relevance > 95% (for 0.01 < *p* value < 0.05).

**Table 3 ijms-24-13359-t003:** Experimental design levels for independent variables used in the synthesis of GelMA.

Factor	Name	Unit	Low Level Value	High Level Value	Level of Variation
X1	Temperature	°C	40	60	5
X2	Methacrylic anhydride over gelatin ratio	%	2.5	152.5	7
X3	Methacrylic anhydride flow rate	mL·min^−1^	1	2.5	7
X4	Stirring speed	rpm	450	650	3

## Data Availability

Not applicable.

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
