# Peer review of "Gelatin Methacryloyl (GelMA) Hydrogel Scaffolds: Predicting Physical Properties Using an Experimental Design Approach"

_ijms, 2023, doi:10.3390/ijms241713359_

Round 1

Reviewer 1 Report (Previous Reviewer 3)

The author has addressed all the comments and improved the quality of the manuscript from the previous version. Thus, I will recommend the manuscript for further publication.

Reviewer 2 Report (Previous Reviewer 1)

The authors satisfactorily answered my comments. To me the paper can be accepted

English is improved but further minor revision is still required. Best if the manuscript is revised by a native english speaker or by a specialized company

This manuscript is a resubmission of an earlier submission. The following is a list of the peer review reports and author responses from that submission.

Round 1

Reviewer 1 Report

The manuscript applies the design of experiments methodology to the study of GelMA hydrogels for tissue engineering applications. Experiments were performed by varying three input parameters and analyzing the response in terms of four output parameters.

The work has some interest, although little is left to scientific innovation. In other words, the reader does not learn very much about the GelMA system and its properties, nor about the applications of GelMA. In summary, it can be considered more like a good technical paper where a methodology is illustrated.

Based on that, the work can be eventually published, but some modifications are in order:

1) in the Introduction, nothing is said about the design of experiments methodology and its applications especially in the area of interest. The authors should insert this point;

2) still in the Introduction, the authors say that “the influence of the methacrylation reaction parameters on the physical properties of GelMA has only been studied using one-factor-at-a-time experimentation, which does not take into consideration the interaction”. The statement is too strong. In the recent literature, some, although few examples, can be found. I ask the authors to correct the statement and to look for and quote the appropriate literature;

3) the Results section is obviously the most relevant one. It is divided into four sub-sections, each one devoted to one of the output parameters. To my opinion, the writing is too heavy and tiring for the reader. Many sentences concerning methodology and data treatment are just the same from one section to the other. Just as a way of example, the statement about the coefficient of determination is essentially repeated (although with slightly different wording) at the beginning of each section. A single general statement, accompanied by a Table with figures, would make the paper shorter and clearer. I ask the authors to review Section 2, by putting at the beginning all general and shared statements about methodology, and limit the subsection to the presentation of the single parameter results only;

4) Section 3, Discussion, is a big mess! The authors change their approach completely by dividing the section into subsections where very general topics, and not the specific parameters actually studied in the work, are discussed. Furthermore, the section mixes together the results of the present work with other results present in the literature, often without a clear and direct correlation and distinction. Just as a way of example, in subsection 3.2 the role of pH on the degree of substitution is presented and discussed by literature references. But the pH is not considered among the parameters studied by the authors! So, what is the reason to introduce it in the discussion? More generally, Section 3 is very confusing, it should be completely rewritten. Maybe the authors could transfer part of the material of Section 3 in the Introduction, leaving in the Section only the direct comparison between the results of their study with those present in the literature and only concerning the same input parameters investigated.

5) English must be improved. Errors, mistakes, misprints are present here and there throughout the manuscript. A deep language revision is required.

Reviewer 2 Report

The article “Gelatin methacryloyl (GelMA) hydrogel scaffolds predicting physical properties using an experimental design approach” presents a thorough study regarding the influence of GelMA synthesis parameters (ratio and flow rate of methacrylate anhydride, temperature, agitation speed) on the resulted properties of the methacryloyl derivative. It is obvious that the authors put a lot of work into their study, developing and characterizing a large number of samples.

I have serious doubts regarding the novelty of this study, since it concludes that the parameter that has the biggest influence on the substitution degree is the amount of methacrylic anhydride – fact that was already established by the literature.

Furthermore, table 1 is very poorly constructed. I suggest a rearrangement of the samples, considering the parameters that were left unchanged. For example – if the second column is temperature, list all samples for 40 degrees C, followed by 50 degrees C, 55 degrees C and then 60. The second column – MA:Gelatin ratio – start for each temperature with the same ratio, and so on. Keep only the samples that may be compared (with only one difference between parameters – if you change two or more parameters at the same time, you can’t have a clear conclusion!).

Standard deviation values should be included for the measured parameters. Samples 21-24 had the same independent variables and resulted (more or less) in the same response – you can sum it up to a single line, adding the standard deviation values.

In addition to these general suggestions, there are small details that can also help better your manuscript.

I would like to make the following suggestions:

1.       Line 17: “since” seems a better choice than “indeed”

2.       Line 21: choose between “to” and “in order to” (you wrote “to in order to”)

3.       Line 23: either “agitation speed during the reaction” or “agitation speed of the reaction mixture

4.       Lines 21-26: the authors should take into consideration that the water uptake capacity, storage modulus, and compression modulus are dictated by the degree of substitution, and they rephrase these lines so that this issue is clear.

5.       Line 32: “adjustable”, not “modular”

6.       Line 60: Methacrylic anhydride is not added, the modification of gelatin is not done through addition.

7.       Lines 128-148: the authors discuss the stirring speed in terms of “lower” and “higher”, without giving any numerical date. I would also like to point out that “rpm” is not a proper way to describe the stirring speed, since this measure depends highly on the diameter of the beaker.

8.       Lines 301-307: the rheological behavior of the hydrogels are given by both storage and loss modulus. While the storage modulus gives information regarding the elasticity of a sample, the loss modulus offers insights about its ability to dissipate energy upon stress. There is no mention about G” in the whole manuscript, although it is depicted in figure 5B.

9.       Lines 359-366: please specify the dimensions of the tested disks

10.   Line 367: what samples are depicted in figure 5?

11.   Line 382: there are several equations in the literature describing the affinity of the hydrogels for aqueous media. Try using one of the equations found in the following articles:

a.       Park H, Guo X, Temenoff JS, Tabata Y, Caplan AI, Kasper FK, Mikos AG. Effect of swelling ratio of injectable hydrogel composites on chondrogenic differentiation of encapsulated rabbit marrow mesenchymal stem cells in vitro. Biomacromolecules. 2009 Mar 9;10(3):541-6. doi: 10.1021/bm801197m. PMID: 19173557; PMCID: PMC2765566.

b.       Thi Sinh Vo, Tran Thi Bich Chau Vo, Trung Tien Tran, Nhan Duy Pham, Enhancement of water absorption capacity and compressibility of hydrogel sponges prepared from gelatin/chitosan matrix with different polyols, Progress in Natural Science: Materials International, Volume 32, Issue 1, 2022, Pages 54-62, ISSN 1002-0071, https://doi.org/10.1016/j.pnsc.2021.10.001

Reviewer 3 Report

In this manuscript, the author has reported GelMA synthesis by varying temperatures, the ratio of methacrylate anhydride/gelatin, methacrylate anhydride addition, flow rate, and stirring speeds. The work is not disclosing any novel synthetic route or a new concept. However, I do not find the novelty of the paper. I think this is not sufficient for publication. However, several experimental findings were compared with previously reported literature which seems very confusing for the reader. I suggest the author to compare the value of several parameters of their current findings with previously reported results in table form. The resolution/quality of all the figures should improve.  Thus, I will not recommend this paper for publication in the International Journal of Molecular Sciences.